# The Canadian Hydrological Model (CHM) v1.0: A multi-scale, multi-extent, variable-complexity hydrological model -- Design and overview

Christopher B. Marsh[1,2]

John W. Pomeroy[1,2]

Howard S. Wheater[2,1]

**Affiliations**: [1] Centre for Hydrology, University of Saskatchewan, Canada.; [2] Global Institute for Water Security, University of Saskatchewan, Canada

## 1 Abstract

Despite debate in the rainfall-runoff hydrology literature about the merits of physics-based and spatially distributed models, substantial work in cold regions hydrology has shown improved predictive capacity by including physics-based process representations, relatively high-resolution semi- and fully-distributed discretizations, and use of physically identifiable parameters that require limited calibration. While there is increasing motivation for modelling at hyper-resolution (< 1 km) and snow-drift resolving scales (~1 m to 100 m), the capabilities of existing cold-region hydrological models are computationally limited at these scales.

Here, a new distributed model, the Canadian Hydrological Model (CHM), is presented. Although designed to be applied generally, it has a focus for application where cold-region processes play a role in hydrology. Key features include the ability to capture spatial heterogeneity in the surface discretization in an efficient manner via variable resolution unstructured meshes; to include multiple process representations; to be able to change, remove, and decouple hydrological process algorithms; to work both at a point and spatially distributed; the ability to scale to multiple spatial extents and scale; and to utilize a variety of forcing fields (boundary and initial conditions). This manuscript focuses on the overall model philosophy and design, and provides a number of cold-region-specific features and examples.

## 2 Key points

- Novel unstructured mesh discretization allows for reduced computational cost while including spatial heterogeneity.

- Ability to modify structure and algorithms within a distributed framework allows for in-depth uncertainty testing.

- Flexible spatial and temporal scales, software abstraction, and robust pre- and post-processing routines allow for incorporating existing code, decreasing development effort.

## 3 Introduction

Hydrological models are important tools for understanding past hydrological events, evaluating anthropogenic impacts on natural systems, and informing water resource and management decisions under contemporary and future climates (DeBeer et al., 2015; Freeze and Harlan, 1969; Milly et al., 2008; Mote et al., 2005; Nazemi et al., 2013; Wheater, 2015). Due to the significant role mountains play in the global water supply as 'water towers' (Viviroli et al., 2007), the fragility of arctic and mountain ecosystems (Bring et al., 2016), and these regions' sensitivity to anthropogenic climate change (Duarte et al., 2012; Mote et al., 2005; Musselman et al., 2017; Rasouli et al., 2015), there is substantial motivation to provide timely and accurate simulations that can be used to address current and future management challenges in these cold regions. Although the need for multi-scale (Samaniego et al., 2017), hyper-resolution (sub-1 km) (Wood et al., 2011), and snow-drift resolving scales (1 m to 100 m) (Pomeroy and Bernhardt, 2017) is becoming clear, contemporary cold-region models suffer from shortcomings when run over large extents and high spatial resolutions and may be limited to what spatial scale they operate at.

Numerous studies suggest that model performance is greatly improved in cold regions when including explicit spatial heterogeneity, identifiable parameter spaces, and a full range of cold regions hydrological processes, e.g., Pomeroy et al. (1998a), Pomeroy et al. (1998b), Bartelt and Lehning (2002), Bowling et al. (2004), Etchevers et al. (2004), Raderschall et al. (2008), Dornes et al. (2008b), Essery et al. (2013), Essery et al. (2009), Pomeroy et al. (2013), Fang et al. (2013), Fiddes and Gruber (2014), Kumar et al. (2013), Endrizzi et al. (2014), Mosier et al. (2016), and Painter et al. (2016). Better understanding of the physical system instead of solely focusing on parameter optimization (Bahremand, 2015) and ensuring that models are not needlessly constrained by rigidity of model structure, choice of parametrization, and representation of spatial variably and hydrological connectivity (Mendoza et al., 2015) are expected to further predictive capacity. Physics-based models may also limit the reliance upon calibrated effective values and decrease uncertainty due to requiring the use of physically-identifiable parameters (Fatichi et al., 2016; Pomeroy et al., 2013). The use of lightly- or uncalibrated models is increasingly important for simulating future conditions as climate non-stationarity increases the uncertainty of calibrated models (Brigode et al., 2013; Vaze et al., 2010). Distributed, physics-based models are thus often the most appropriate type of hydrological model for simulating distributed state variables (Dornes et al., 2008a; Fatichi et al., 2016), simulating catchments with extreme heterogeneity (Kumar et al., 2013), or when simulating process interactions (Dornes et al., 2008a; Horne and Kavvas, 1997; Maxwell and Kollet, 2008). These improvements motivate the continued development of spatially discrete, physics-based models.

Although there is uncertainty in the optimum levels of complexity required in cold region hydrological models (Avanzi et al., 2016; Clark et al., 2017), such models have unique requirements and considerations; a brief summary follows. The largest discharge event of the year often results from the melt of the seasonal snowpack (Davies et al., 1987; Gray and Male, 1981) and therefore substantial effort has been invested in snow model development, e.g., Jordan (1991), Marks et al. (1998), Bartelt and Lehning (2002), Vionnet et al. (2012), Leroux and Pomeroy (2017), and flexible snowcover modelling systems, e.g., the Factorial Snow Model (FSM) (Essery, 2015) and ES-CROC (Ensemble System Crocus) (Lafaysse et al., 2017). Streamflow discharge is impacted by snowmelt spatial heterogeneity that is due to: variability in surface energetics (Carey and Woo, 1998; Dozier and Frew, 1990; Harder et al., 2019; Marks et al., 1992; Mott et al., 2013; Munro and Young, 1982; Olyphant, 1986; Pomeroy et al., 2003; Schlögl et al., 2018), precipitation spatial variability (Harder and Pomeroy, 2013; Lehning et al., 2008; Marks et al., 2013), vegetation canopy interception (Hedstrom and Pomeroy, 1998; Kuchment and Gelfan, 2004), and snow redistribution via wind processes (Essery et al., 1999; MacDonald et al., 2009; Mott et al., 2010; Pomeroy et al., 1993; Pomeroy and Li, 2000; Winstral et al., 2002). Snowmelt runoff is further complicated due to frozen soils that limit infiltration rates (McCauley et al., 2002; Zhao and Gray, 1999) such that standard infiltration representations are insufficient (Lundberg et al., 2016). Active layer depth above permafrost dramatically impacts surface characteristics (e.g., topography, vegetation, soils), streamflow seasonality, and water partitioning (Walvoord and Kurylyk, 2016). In cold regions, the numerous lakes and wetlands impact the local climate during ice-free periods (Latifovic and Pouliot, 2007; Rouse et al., 2005; Shook et al., 2015). In summary, cold regions hydrological models have unique challenges and must include a variety of process representations not considered in most temperate hydrological models.

There are, however, significant limitations in hydrological modelling ability. For instance, there are deficiencies due to substantial heterogeneity and difficulty in observing surface and subsurface parameters and processes (Freeze, 1974), no single scale at which homogeneity of control volumes is achieved (Beven, 1989; Blöschl and Sivapalan, 1995; Klemeš, 1983; Shook and Gray, 1996), and mismatches between underlying theory and applied scales (Or et al., 2015). These limitations manifest as 1) uncertainties in model parameters, initial conditions, boundary conditions, forcing data; 2) incomplete process representations, selections, and linkages (Beven, 1993; Beven and Westerberg, 2011; Clark et al., 2008; Fatichi et al., 2016; Raleigh et al., 2015; Slater et al., 2013; Wagener and Montanari, 2011); and 3) issues of complexity including the degree of physics-based equations, the number of parameters, forcing data requirements, and spatial discretization requirements (Beven, 1993; Clark et al., 2008; Hrachowitz and Clark, 2017). In addition, these models may have limited structural flexibility for incorporating multiple modelling philosophies (e.g., Dornes et al. (2008b), Clark et al. (2011)), or have limitations in incorporating next-generation data products such as unmanned aerial vehicle (UAV) imagery (Bühler et al., 2016; Harder et al., 2016; Spence and Mengistu, 2016). Without care, physically based, mechanistic approaches can result in over parameterized models (Perrin et al., 2001) that are highly uncertain and difficult to verify (Beven, 1993) due to a mismatch in model element and observed scales and limited high-resolution spatially distributed data

(Beven, 1989). Physically based models should be used critically, with proper appreciation of the strengths and the limitations, and dependent on the purpose of the modelling (Beven, 1993, 2006; Das et al., 2008; Perrin et al., 2001).

In order to address the scientific and societal demands placed on hydrological models, there is a need for a new generation of hydrological models that allow:

5    1.    *Multi-scale, spatially distributed process representation*

Although semi-distributed schemes such as the Group Response Unit (GRU) or Hydrological Response Unit (HRU) approach have had substantial success in cold regions, e.g., Pietroniro et al. (2007), Pomeroy et al. (2007), and Clark et al. (2015), complex spatial behaviours cannot be modelled unless the HRUs are constructed *a priori* to produce the behaviours. This limits simulating cascading processes and emergent behaviours, e.g., accumulation of non-linear process interactions leading to basin-wide behaviours. Representing mass and energy heterogeneities and interactions, at multiple spatial scales (Hrachowitz and Clark, 2017; Samaniego et al., 2017), and moving towards regional predictions (Sivapalan, 2017) has been suggested as a path to improving predictive capacity. Fully distributed, raster-based models are inefficient with the need for many raster cells, greatly limiting the applicability for both high resolution, and over large extents. The deficiencies in HRU, GRU, and raster-based models points towards a need for an improved terrain representation that allows both high resolution as needed and applicability for modelling over large extents.

2.    *Flexible model structure*

Many models use a rigid model structure that does not allow for easily changing model algorithms and all parameters nor easily testing different algorithms or hypotheses. An improved approach is to allow process modularity for easily modifying aspects of a model's structure and complexity. Such model flexibility has been present in many rainfall-runoff models, e.g., MMS (Leavesley et al., 2002), FUSE (Clark et al., 2008), SUPERFLEX (Fenicia et al., 2011), but to the authors' knowledge, such modularity in cold-region models has been limited to the Cold Regions Hydrological Model (CRHM) (Pomeroy et al., 2007) and SUMMA (Clark et al., 2015). These are both modular, physics-based, semi-distributed hydrological response unit (HRU) models with capability for cold regions hydrology. A flexible model structure should allow for easily scaling between temporal scales (i.e., time-stepping), spatial extents, spatial resolutions, and process representations as required. Assumptions on explicit coupling between processes leads to difficulty in testing different process representations and limits inclusion of existing code. Despite the rich set of cold regions snow and hydrological models, e.g., Alpine3D (Lehning et al., 2006), iSnobal (Marks et al., 1998), GeoTOP (Endrizzi et al., 2014), MESH (Pietroniro et al., 2007), CRHM (Pomeroy et al., 2007), SUMMA (Clark et al., 2015), SRGM (Gelfan et al., 2004; Kuchment and Gelfan, 2004), ESCROC (Lafaysse et al., 2017), and VIC (Cherkauer et al., 2003), there are no explicitly distributed, modular cold regions models.

3.  *Ease of changing model parameters, initial/boundary conditions*

Model parameters, initial conditions, and boundary conditions are uncertain in hydrological systems and are a significant constraint on model complexity and validity. Hard-coded parameters can be a significant source of uncertainty as they are effectively treated as physical constants (Mendoza et al., 2015). Modern models must be developed so that changing initial conditions, parameters, and all aspects of the model configuration are trivial and easily done within the context of an uncertainty framework. Due to the long temporal durations for which climate change scenarios are done, flexibility in changing surface parameters with time, e.g., vegetation cover, needs to also be possible.

4.  *Efficient use of computational resources*

Unlike GRU or HRU based models, distributed models are generally discretized using a raster approach with a fixed spatial resolution. This can lead to either increased computational requirements or non-optimum use of computer resources due to the over representation of the surface (e.g., homogenous locations), while choosing a coarser sized mesh may result in failure to capture quickly varying, and extremely important heterogeneity. Because of the general over representation of topography via a fixed-resolution raster, these distributed models become difficult to parametrize, and computationally expensive to run, limiting their applicability to large spatial extents. Using more efficient terrain representations as well as modern high-performance computing paradigms can reduce this wasted computational effort.

5.  *Allow appropriate model complexity*

Raster-based models with high resolution grid cells, and wasted computational effort as noted above, often led to arbitrary complexity reduction and process removal due to computational constraints. Reducing the model runtime is often a justification for simpler conceptual models, for simpler landscape representations, and for fewer computational elements. Hydrological model complexity should be warranted based upon the simulation results and needs and not for simplicities sake.

Although there are substantial advantages to the benefits of using physics-based, fully distributed models, data (forcing and validation) and computational limitations that have slowed their development and adoption. However, recent technological progress has been progressively removing some of these limitations. For example, unmanned aerial vehicle (UAV) imagery is providing sub-metre digital surface and elevation maps (Bühler et al., 2016; Harder et al., 2016), vegetation classification (Spence and Mengistu, 2016), hydrological features (Spence and Mengistu, 2016), as well as initial conditions, e.g., snowcover (Bühler et al., 2016; Harder et al., 2016). Surface geophysical methods are improving characterization of large-scale subsurface properties (Hubbard et al., 2013). Remote sensing products of soil properties are of increasingly higher quality (Mohanty, 2013), and high resolution satellite imagery can be used to diagnose spatial patterns of snowcover

(Wayand et al., 2018). Wide-spread access to High Performance Computing (HPC) resources, e.g., Compute Canada [Canada], Extreme Science and Engineering Discovery Environment (XSEDE) [United States], National Computational Infrastructure (NCI) [Australia], Horizon2020 initiative [European Union], can help offset the increased computational cost of the simulations and of the uncertainty analysis needed to constrain *a priori* estimated physically-based parameters (Paniconi and Putti, 2015). Lastly, efficient uncertainty analysis frameworks such as VARS (Razavi and Gupta, 2016), can also decrease the total number of required simulations to estimate uncertainty, further reducing the computational burden. However, estimates of critical subsurface properties such as hydraulic conductivity cannot be represented a priori with sufficient confidence, nor at the correct scale, viz. effective model element parameters (Binley et al., 1989), to avoid calibration (Freeze, 1974).

In summary, models will always require a trade-off between computational complexity (e.g., algorithms, landscape representation, initial conditions, parameters, and terrain discretization) and model performance (e.g., modelled versus observed). Cold region hydrological models have unique requirements that motivate the inclusion of explicit spatial heterogeneity via semi and fully distributed discretizations. To simulate the complex inter-process interactions that lead to important hydrological features, a variety of features must exist within a distributed, process-based modelling framework.

This manuscript outlines the philosophy and details of a new hydrological model, the Canadian Hydrological Model (CHM), and how the development of this modelling framework addresses the above outlined limitations of many existing hydrological models and contributes to cold regions modelling. This manuscript focuses on the overall model philosophy and design, and provides a small number of cold-region specific features and examples.

## 4 Design and Overview

### 4.1 Overview

The Canadian Hydrological Model (CHM) is a spatially distributed, modular modelling framework. Although not restricted to cold regions, it is designed with both cold regions and temperate zone processes in mind and has various capabilities that facilitate the modelling of these domains. The design goal of CHM is to use existing high quality open source libraries and modern high-performance computing (HPC) paradigms. By providing a framework that allows for as loose or tight a coupling between processes as required, CHM allows integration of current state-of-the-art process representations and makes no assumptions about the complexity of these process representations. For example it allows testing of the representations in a consistent manner, diagnosing model behaviour due to parameter changes, process representation changes, and basin discretization. Spatially, it allows for domains at point ($10^{-6}$ km$^2$), hillslope (1 km$^2$ to 10 km$^2$), basin (100 km$^2$), regional (8,000 km$^2$), and provincial/state (> 1,000,000 km$^2$) scales. The following sections outline the framework features, including terrain representation, surface parameterization, process representation, meteorological inputs,

parallelism, uncertainty analysis, visualization and analysis, and adaptation of raster algorithms. Although the CHM will eventually include the entirety of the hydrological cycle, at this time only snow accumulation and surface meteorology processes are implemented. Additional model components are being developed, and will be available in future versions of CHM.

## 4.2 Terrain representation

The spatial variability of terrain is a key component to any model and is an important component of model complexity. Regardless of how sophisticated, physics-based, and spatially explicit a hydrological model may be, at some level the hydrological system is conceptualized and aggregated into a control volume (Vrugt et al., 2008). Structured meshes, also known as rasters and grids, are a landscape discretization where the landscape is discretized by uniform sized cells. Raster-based hydrological models are common (Tucker, 2001) because their computer representation is trivial, and widespread use of rasters, such as in remote-sensed data, makes using them a natural choice in hydrological models. However, rasters have a number of significant limitations, the most limiting being a fixed spatial resolution over the entire basin (Tucker, 2001). This results in potentially large computational inefficiencies due to over-representation of topography. This arises as a result of requiring small raster cells (elements) to capture the spatial variability in areas of high topographic variability or (sub-) surface variability (e.g., vegetation, soils), which results in over-representation of areas that have limited spatial variability. Coarse resolution rasters also have discontinuities in the elevation data, where adjacent cells may have large elevation differences.

Unstructured triangular meshes, sometimes referred to as Triangulated Irregular Networks (TINs), represent the topography via a set of irregularly sized, non-overlapping connected triangles, where each triangle face is of a constant slope (Chang, 2008). Areas of large topographic variability can have a higher density of small triangles in order to capture the spatial variability and areas of relatively homogeneous topography have fewer large triangles. This a more efficient terrain representation than rasters (Shewchuk, 1996), and may have up to a 90% reduction in computation elements (Ivanov et al., 2004; Marsh et al., 2018). Despite these computational advantages, a practical downside is that due to the widespread availability of raster data, conversion to an unstructured mesh is required. This results in increased uncertainty due to aggregation of the landscape into control volumes. The CHM uses a novel multi-objective approach for unstructured triangular mesh generation, *Mesher*, detailed in Marsh et al. (2018). A brief summary follows: quality Delaunay meshes are generated ensuring a smooth graduation between small and large triangles; triangles are bounded with minimum and maximum triangle areas to ensure process representations match the physical scale; triangles are generated to fulfil tolerances (e.g., RMSE) to the underlying topographic raster and other important landscape features such as vegetation and soils. This mesh generation attempts to limit the amount of error introduced by the approximating surface given by the unstructured mesh and provide mechanisms to ensure spatial heterogeneity in the landscape is correctly preserved.

Using this mesh generation, simulation domains can be constructed at a variety of spatial extents, and importantly, spatial scales. An example of this variable resolution triangulation mesh for a part of the Bow River Basin in the Canadian Rockies and foothills west of Calgary, Alberta, Canada is shown in Figure 1. The triangular edges are shown in grey lines. The variable resolution produces larger triangles in the valley bottoms, where topographic variability is limited, and small triangles in the mountains, where the heterogeneity is greater. This allows for diagnosing the impact of scale on model performance as well as matching the process representation to the correct model length scale. Further constraints could ensure streams are accurately defined.

### 4.3 Triangle parameterization

Setting values of parameters for the triangles, such as assigning vegetation or soil type to the triangle, is done during the mesh generation phase. The parameter values are stored in a file separate from the underlying mesh, and thus can be easily changed at run time. This allows for easily investigating the impact of parameter values on outputs. The parameterization of the triangles is done by a) determining the valid raster cells under each triangle and b) calculating an error metric for these cells and assigning this value to the triangle. Maximum and mean are the two most commonly used methods, but it can be any user-defined function. For classified data, the mode is used. This would allow, for example, selection of the most dominant landcover class. In addition, a user-specified classifier function can be given to easily classify continuous input parameters; e.g., classifying vegetation-heights into vegetation classes. Lastly, CHM provides mechanisms to write model output to a format that can be used as input; that is, CHM can use its output to set triangle values for future simulations.

### 4.4 Modular process representation structure

A hydrological model is a hypothesis based on assumptions of how a hydrological system works (Savenije, 2009). Modular model structures allow for rigorously testing process representations and have been used with success in cold regions hydrology, e.g., Cold Regions Hydrological Model (CRHM) (Pomeroy et al., 2007), and Structure for Unifying Multiple Modeling Alternatives (SUMMA) (Clark et al., 2015). A feature of CHM is that it provides a modular process representation that is suitable for distributed modelling, while maintaining high computational performance and flexibility.

In CHM, process representations are conceptualized into *modules*. Selecting various combinations of these modules in the CHM framework defines the overall model. A principal design goal of the module system is that a module has an enforced set of pre- and post-conditions. Pre-conditions represent the variables that must be computed prior to a given module running, and post-conditions encapsulate variables that must be computed by the currently running module so-as to be available as input for other modules. At run time, the user-selected set of modules are linked together into a directed acyclic graph based on these variable dependencies, and module execution order is determined via a topological sort of this graph. This sort ensures that modules are run in an order so-as to fulfil the pre-condition (i.e., the variable dependencies). Linkages between modules showing these dependencies are shown in Figure 2. The lines with arrows show how variable dependencies

are resolved between modules. The lines going from a module are the post-conditions that satisfy the pre-conditions of the next-to-be-run module. In this example, a snowcover model, Snobal, is being driven by meteorology with the output of Snobal being used as input to a frozen soil infiltration model (Gray_inf).

The hydrological literature has a diverse set of process representations that are either one-dimensional with no lateral exchange between elements (point-scale) or are explicitly coupled with surrounding elements (Todini, 1988). CHM makes no assumption about either, and modules may either operate on a single triangle, or on the entire domain. If only point-scale modules are selected, then CHM may be optionally run at a point-scale, effectively disabling the rest of the distributed framework. As there are substantial merits to mixing top-down and bottom-up process representations (Hrachowitz and Clark, 2017; Pomeroy et al., 2004), CHM makes no assumptions on the complexity or type of process representation in a module – modules may be a mix of complex physics based representations and conceptual representations. This also applies to process coupling. For example, a module could be a single process (e.g., a snow model), a coupled set of processes (e.g., coupled heat and energy snowmodel + frozen soil routine), or an entire existing model.

Due to the strict pre- and post-conditions required for module dependency resolution and the abstraction used in CHM, existing libraries and code can be used in a model. There is no need to rewrite the code. Therefore, any code that may be called via a C interface (e.g., Fortran, R, Python, Matlab) is suitable to be used (with a few considerations) as a CHM module.

Summarized in Table 1, and described in brief below, are a list of the processes currently available in CHM. Two energy balance snowpack models are available, Snobal and SNOWPACK. Snobal (Marks et al., 1999) is a two-layer energy balance model with a fixed upper layer that is used for the estimation of outgoing longwave radiation and atmosphere-snow temperature gradients for the turbulent heat flux. SNOWPACK (Bartelt and Lehning, 2002) is a multi-layer finite element energy balance snow model originally developed for avalanche hazard forecasting. In addition to the snowcover albedo estimates provided by Snobal and SNOWPACK, the Canadian Land Surface Scheme (CLASS) albedo routine is available (Verseghy, 1991). Frozen soil infiltration is calculated using the parametric form of Gray et al. (2001). Horizontal snow mass is redistributed using a 3D advection-diffusion blowing snow model derived for unstructured meshes (Marsh et al. (2019), in review). Blowing snow saltation (Pomeroy and Gray, 1990), turbulent suspension (Pomeroy and Male, 1992), sublimation (Pomeroy et al., 1993), threshold shear stress for saltation (Li and Pomeroy, 1997), shear stress partitioning by vegetation and snow, and probabilistic upscaling (Pomeroy and Li, 2000) parameterizations comprise the blowing snow model. Vertical redistribution of mass in steeply sloping terrain is calculated using Snowslide (Bernhardt and Schulz, 2010) using a threshold slope and mass exceedance to transport mass downslope (i.e., it is not a prognostic avalanche model). The forest canopy is conceptualized into open and forest areas and uses the snow interception algorithm of Hedstrom and Pomeroy (1998) coupled to the intercepted snow sublimation and unloading algorithms of Pomeroy et al. (1998b) and the drip and rapid

unloading formulations in Ellis et al. (2010). Sub-canopy short and longwave irradiance and turbulent transfer algorithms from Ellis and Pomeroy (2007), Pomeroy et al. (2009) and Ellis et al. (2010) are also included in the canopy module.

## 4.5 Input meteorology

Input meteorology is prescribed as a point source (herein, 'virtual station') defined by latitude, longitude, and elevation. However, a virtual station may have an arbitrary location and elevation and need not be within the simulation domain, nor correspond to a real meteorological station. This allows a virtual station to be located at, for example, the centroid of a numerical weather prediction output grid cell. Because all input meteorology is given as a point source, various spatial interpolants are present in CHM to provide a distributed field across all triangles.

Spatial interpolates are present as inverse distance weighting (IDW) and thin plate spine with tension. In some cases, no interpolation is desired, and therefore a third option called 'nearest' is available – this uses the nearest virtual station without any spatial interpolation. Over large domains, such as when using numerical weather prediction output, every virtual station in the simulation domain should not be used in the interpolation to every triangle. Therefore, interpolates may query a list of either: a) virtual stations within some distance of the triangle or b) the closest $n$ virtual stations. This ensures that only nearby virtual stations are used to form the interpolant. Vertical elevation correction is provided by a set of specialty modules. All virtual stations are corrected to a common reference level using these modules prior to spatial interpolation. A list of these algorithms is summarized in Table 2.

Input meteorology may be given as either text files or as NetCDF files (Rew and Davis, 1990). When NetCDF files are used, the timesteps' data are lazy-loaded such that only the current timestep is read. This decreases the up-front load time as well as decreases total memory usage.

## 4.6 Input filters

Input filters provide a mechanism to modify input meteorology during runtime. This is similar to the filter feature in CRHM (Pomeroy et al., 2007) and MeteoIO (Bavay and Egger, 2014). Filters are assigned to each virtual station, and each virtual station may have an arbitrary number of filters. The purpose of filters is to allow, for example, values outside of a certain range to be filtered, or to perform a correction such as taking an observed wind speed at 2 m and changing it to 10 m for use later in a process module. Filters operate per-timestep and therefore can consider the previous model timestep for use in the correction; e.g., including snow depth to perform vertical wind speed height correction.

## 4.7 Point mode

Due to the difficulty in validating spatial models due to limited spatial observations, evaluation is generally performed using point observations. CHM may be run in point-mode that allows for simulating a single triangle without lateral interactions,

using a specialized input module to pass a hydrometeorological station's observation data directly to the underlying process models. This is intended to simulate a point collocated with an input observation meteorology dataset and allows for traditional point simulations.

### 4.8 High Performance Computing

In CHM, parallelism is currently implemented via the shared memory OpenMP library. Coding a process representation into a module will generally result in either a point-scale module (e.g., point-scale snowcover model) or it will be a spatially coupled model (coupled advection-diffusion equation). The first type, owing to the fact it does not require knowledge of its neighbours to compute a value, corresponds to an embarrassingly parallel problem – that is, a problem that does not require any communication between threads. Herein, these are referred to as data parallel. Spatially coupled models require the

solution at their neighbour triangles in order to compute a solution. These neighbours, in turn, require solutions at *their* neighbours, and so on. Therefore, this is a much more challenging type of problem to introduce parallelism to. Herein, these are referred to as domain parallel. Data parallel modules automatically have the parallelism implemented and require no special consideration from the developer. Domain parallel modules, however, require the module developer to implement parallelism as appropriate for the module.

Mixing these two types of parallelism complicates the implementation of parallel code. To provide as much seamless parallelism as possible, each module declares the type of algorithm it is: data parallel or domain parallel. After the topological sort is performed to determine module execution order, the modules are scheduled together into groups that share a parallelism type. For example, consider the following sorted list of modules, with their parallelism type in brackets:

mod_A (parallel::data)

mod_B (parallel::data)

mod_C (parallel::data)

mod_D (parallel::domain)

mod_E (parallel::data)

These would then be scheduled together into 3 groups:

**Group 1**

mod_A (parallel::data)

mod_B (parallel::data)

mod_C (parallel::data)

**Group 2**

mod_D (parallel::domain)

**Group 3**

mod_E (parallel::data)

The modules in group 1 are run in parallel together. Because they are data parallel, only one iteration over the mesh is required. Then, groups 2 and 3 are run. This scheduling mechanism reduces the overhead of a modular approach by limiting total iterations over the mesh and minimizing thread creation. Further, as most hydrological process representations are point-scale, it allows for abstracting parallelism, resulting in "free" parallelism for the developer.

### 4.9 Uncertainty analysis

CHM provides a mechanism to easily allow modules to obtain parameter values from configuration files (JSON format), overriding the default hard-coded value. Changes to the model structure (i.e., choosing modules), initial conditions, and parameter files (e.g., landcover) are also done via this mechanism. Users may, via the command line, change any configuration value – thus simplifying uncertainty testing. This mechanism reduces situations were changes require re-compilation.

The Python code snippet shown in Listing 1 demonstrates changing values on the command line (via Python). This code is setting the name of three output files and adding a new module to be run.

**Listing 1: Example to setting output file names and adding a new module.**

```python
import subprocess
import shutil

prj_path = "CHM.config"

cf1 = "-c output.VistaView.file:vv_dodson.txt"
cf2 = "-c output.UpperClearing.file:uc_dodson.txt"
cf3 = "-c output.FiserraRidge.file:fr_dodson.txt"
cf4 = "--add-module Dodson_NSA_ta"

subprocess.check_call(['./CHM %s %s %s %s %s' % (prj_path, cf1, cf2, cf3,cf4)], shell=True
```

## 4.10 Visualization and analysis

The output format used is the ParaView (Ahrens et al., 2005) unstructured mesh format. This allows for visualization of the simulation results in full 3D, with timeseries analysis in ParaView, as shown in Figure 3. The addition of a ParaView plugin for CHM allows for displaying the date and time of the output. The animation view allows for exploring the spatio-temporal results. It also allows for immediate diagnosis of modelling errors, especially if the spatial pattern of an output variable is clearly incorrect. For example, if a coding error resulted in: a patch-work of air temperatures instead of an expectedly smooth gradient with elevation, snowdrifts being formed in locations that were known to be incorrect such as the top of a ridge instead of in the lee, or northern hemisphere north-facing slopes receiving the most shortwave irradiance. There are many post-processing filters and tools available in ParaView, such as plotting an individual triangle's values over time. Because ParaView uses the Visualization Toolkit (VTK) library (Schroeder et al., 2006), the ParaView files can easily be loaded and post-processed using the Python VTK library in conjunction with traditional Python libraries such as NumPy (Oliphant, 2006) and SciPy (Jones et al., 2018).

In addition to the ParaView output, CHM provides a set of post-processing scripts that allows for converting the Paraview file to a rasterized GeoTiff or NetCDF file. This allows for using the output in post-processing algorithms that require arrays, or in GIS.

## 4.11 Adaptation of raster-based algorithms

Adaptation of raster-based algorithms is an important aspect of CHM as many existing algorithms are raster-based. Frequently, raster-based algorithms employ logic that performs queries such "look X length units in direction Y". This is easily done on a structured mesh, however on an unstructured grid, this process is non-obvious. Iterating over each triangles' neighbours results in a random walk across the domain, and brute-force iteration search methods are needlessly slow. CHM uses the $k$-d spatial search tree available within the dD Spatial Searching (Tangelder and Fabri, 2018) package in the Computational Geometry Algorithms Library (CGAL) to optimize spatial queries. Briefly, a $k$-d tree is a generalization of a binary search tree in high dimensions that decomposes the search domain into a set of small sub-domains (Bentley, 1975). This tree structure can then be reclusively searched resulting for efficient spatial look-ups. The $k$-d tree implementation is how nearby stations are determined. This technique for spatial searching can also be used to calculate terrain parameters, such as the terrain curvature.

## 5 Model application

### 5.1 Overview

The following section describes the methodology for evaluating various features of CHM as well as providing examples of usage. Although the CHM will eventually include the entirety of the hydrological cycle, snow accumulation and surface meteorology processes are currently implemented. Marmot Creek Research Basin (MCRB) in the Canadian Rockies in Alberta, Canada is used as a location to test the two snow modules and various models to provide the driving meteorological forcing for these models that are currently implemented in CHM. The meteorological interpolants are tested in a leave-one-out validation across the MCRB. In addition, an adaptation of a raster-based terrain-shadowing for shortwave irradiance calculation is presented, demonstrating the conversion of an algorithm from a raster to unstructured mesh. Finally, the parallel computation aspect of CHM is tested by performing a scaling analysis using different number of CPUs.

### 5.2 Study Site

#### 5.2.1 Marmot Creek

Marmot Creek Research Basin (MCRB) (Golding, 1970) is located in the Kananaskis River Valley of the Canadian Rockies, as shown in Figure 4. It is a 9.4 km² basin covered predominately by needle-leaf forest (Fang et al., 2013; Pomeroy et al., 2012). The climate is dominated by continental air masses with long and cold winters; however these are interrupted by frequent chinooks (Foehns) in mid-winter (DeBeer and Pomeroy, 2009). It spans an elevation range from 1700 m to 2886 m (Rothwell et al., 2016) and snow covers the upper elevations of the basin from October to June. The average seasonal precipitation is approximately 600 mm at low elevations increasing to over 1140 mm at the tree line (Rothwell et al., 2016).

#### 5.2.2 Meteorological observations

Meteorological observations for air temperature, relative humidity, wind speed, precipitation, soil temperature, and incoming shortwave radiation for the Upper Clearing site (1860 m), Vista View (1956 m), and Fisera Ridge (2325 m) sites, shown as crosses in Figure 4, were used. Gap-filled, quality-corrected 15-min data for the water years 2007 to 2016 (inclusive) were used. Please see Fang et al. (2019) for further details. Precipitation was measured with Alter-shielded Geonor weighing precipitation gauges and corrected for wind-induced under-catch (Smith, 2009). Precipitation phase was determined via the psychrometric energy balance method of Harder and Pomeroy (2013). Longwave irradiance was calculated following Sicart et al. (2006). This was developed for mountainous terrain and was shown to have an error of less than 10% over the snowmelt season. This method has been used with success at the MCRB.

Periodic snow surveys of depth and SWE on long transects at Upper Clearing were conducted by various members of the Centre for Hydrology and used to quantify snowpack density. For each transect, there were at least 25 snow depth measurements and at least 6 gravimetric snow density measurements using an ESC-30 snow tube (Fang et al., 2019).

### 5.3 Models

#### 5.3.1 Snow models

Point-scale evaluation of the two snow models in CHM, Snobal and SNOWPACK, was done at the Upper Clearing site.

Snobal (Marks et al., 1999) is a physics-based, two layer snowpack model designed specifically for deep mountain
snowpacks and approximates the snowpack by two-layers where the surface fixed-thickness active layer (taken here as 0.1 m) is used to estimate surface temperature for outgoing longwave radiation and atmosphere-snow exchange of sensible and latent heat via turbulent transfer. Snobal features a coupled energy and mass balance, internal energy tracking, and liquid water storage calculations. Turbulent fluxes are explicitly calculated via Marks et al. (1992), a bulk transfer approach that includes a Monin-Obukhov stability correction. The ground heat flux is calculated from conduction with a single soil layer
of known temperature.

SNOWPACK (Bartelt and Lehning, 2002; Lehning et al., 2002) is a multi-layer finite element model of mountain snowpacks, with application for avalanche hazard forecasting. It describes the microphysical properties of a snowpack and includes the dynamic addition/removal of snow layers using a system of PDEs. These are discretized vertically into an arbitrary number of snow layers in a Lagrangian coordinate system. It has a coupled energy and mass balance, internal
energy, and liquid water storage calculations with a bulk-transfer turbulent flux scheme with Monin-Obukhov stability correction (Michlmayr et al., 2008). The default Michlmayr et al. (2008) scheme was used herein.

Both SNOWPACK and Snobal were configured to use the albedo routine of Verseghy et al. (1993). The snow models were driven with observed precipitation, shortwave irradiance, wind speed, air temperature, relative humidity, and soil temperature at a 15-minute time interval. Because of the sheltered nature of the Upper Clearing, no blowing snow was
simulated (Musselman et al., 2015). Snow model parameters, such as roughness length, were set following Pomeroy et al. (2012).

#### 5.3.2 Mesh generation

The unstructured mesh was created using the *Mesher* software. A 1 m x 1 m input elevation LiDAR DEM (Hopkinson et al., 2011) was used. The resulting mesh was generated to have a minimum triangle area equivalent to a 25 m x 25 m raster and
represented the topography to within 25 m RMSE. This resulted in approximately $\approx$ 100,000 triangles.

#### 5.3.3 Raster algorithm adaptation (shadowing)

An example of the adaptation of a raster algorithm to the unstructured mesh is shown for a terrain shadowing algorithm, illustrated in Figure 5, that calculates the shadows cast from surrounding terrain. The "look X length units in Y direction" query is required for finding obstructing terrain (e.g., a tall mountain) by searching along the azimuthal direction towards the

sun. As a demonstration of the *k*-d tree usage in CHM, the shadowing algorithm of Dozier and Frew (1990) (herein DF90) was implemented for unstructured triangular meshes. In brief, the DF90 algorithm searches along an azimuthal direction within some horizontal distance and attempts to find terrain that is above the solar elevation. For an observer *A*, a search along the azimuth that corresponds to the solar vector *S* is performed. For each terrain element found, a new vector (*H*) is calculated. If the slope of *H* is greater than that of S, *A* is in shadow. Terrain is searched from the observer towards some maximum search radius, in steps of size *dx*. Specifically, this adaptation of DF90 required using the *k*-d tree to find the triangle at a distance *X* m from the source triangle (A) along an azimuth that corresponded to the solar vector, S.

The DF90 shadowing algorithm was run for all of Marmot Creek, using the mesh described in Section 3.2. A maximum search radius of 1000 m was used, discretized into 10 steps. The guidelines for choosing these search values follows two criteria: 1) the radius should be large enough to cover the distance across a representative valley length distance, such that shadows from mountains across the valley are included; and 2) the step should be about half of a triangle length scale such that steps do not pass over triangles. The DF90 implementation was compared to: observed shadowed area (see below), the Marsh et al. (2012) shadowing model, and the Solar Analyst (Fu and Rich, 1999) shadow model. Solar Analyst is an extension in the ArcGIS software by Environmental Systems Research Institute (ESRI). The observed shadowed area are from time-series images from the field campaign detailed in Marsh et al. (2012) and were orthorectified using the software of Corripio (2004). Shadow location for February 1, 2011 at 17h00 was used in this comparison. The output from CHM was rasterized from the unstructured mesh at a 1 m x 1 m spatial resolution.

**5.4 Leave one out comparison**

To test the efficacy of the meteorological interpolates, a leave-one-out comparison was conducted for the Upper Clearing, Vista View, and Fisera Ridge stations. This entailed using two of the three meteorological stations as input for CHM, in order to predict the third. For example: Upper Clearing and Vista View were used to predict meteorological conditions at Fisera Ridge; Vista View and Fisera Ridge were used to predict Upper Clearing; et cetera.

Ten water years using 15-minute data were simulated. The following meteorological interpolants were used: terrain shadowing (Dozier and Frew, 1990), cloud fraction (Walcek, 1994), air temperature (Cullen and Marshall, 2011), relative humidity (Kunkel, 1989), precipitation phase (Harder and Pomeroy, 2013), precipitation (Thornton et al., 1997), and solar radiation transmittance estimated from observed incoming shortwave values.

**5.5 Parallel scaling**

The heterogenous Westgrid cluster Graham was used to investigate the scaling performance of the CHM code with various numbers of CPUs. The base nodes were used. These have two Intel E5-2683 v4 Broadwell CPUs at 2.1Ghz for a total of 32

cores and 128GB of RAM. The modules run include the data parallel Snobal snowpack module, as well as a domain parallel advection-diffusion blowing snow module (Marsh et al., 2019 in review).

Simulations were run for a mesh with ≈ 100,000 triangles. The model was run with 1, 2, 4, 6, 8, 16, and 32 cores, and for each core-count scenario, the fastest of 5 runs was taken. File output was disabled for these runs. The speedup for the $n$ core run ($core_n$) was computed relative to the 1-core run ($core_1$):

$$speedup = \frac{core_1}{core_n}. \qquad (1)$$

## 6 Results

### 6.1 Point scale snowmodel

Shown in Figure 6 is the simulated snow water equivalent (SWE [mm]) for SNOWPACK (blue) and Snobal (red). The water year is denoted above each plot. Snow course observations are shown as black dots. The RMSE and MB values for both models, for each water year, are shown in Table 3 and averaged over all years in Table 4.

In 2007, SNOWPACK over estimates peak SWE more than Snobal, although ablation timing between the two is identical. In 2008, early season SWE is over estimated by SNOWPACK although late season SWE is better estimated by SNOWPACK. Water year 2009 is poorly simulated in general, especially by Snobal. It is not clear what causes this poor performance. During the cold winters of 2010 and 2011, both models perform well. In 2012, Snobal underestimates peak SWE versus SNOWPACK. For years 2013 to 2015 SNOWPACK better captures peak snow and the ablation period than Snobal. In 2016 Snobal better estimates SWE as SNOWPACK overestimates during accumulation and for peak SWE. SNOWPACK tends to be more consistent in its prediction capacity, although it tends to over estimate, whereas Snobal tends to underestimate total SWE. Overall SNOWPACK tends to perform better than Snobal, although there are individual years where Snobal edges out SNOWPACK.

### 6.2 Adaptation of raster-based algorithm

Shortwave irradiance corrected for slope and aspect, with horizon (cast) shadows via an adaptation of the Dozier and Frew (1990) shadowing algorithm for unstructured meshes for the Marmot Creek Research Basin is shown in Figure 7. Simulation is for 2011-02-01 17:00 local time. High irradiance is shown in red, and shadows shown in dark blue; and these areas are receiving only diffuse radiation. The region north of Fisera Ridge is shown in detail in Figure 8. This figure shows an orthorectified terrestrial photo of a shadow passing over Mt. Collembola from Fisera Ridge. The location of the shadowed region for 2011-02-01 17:00 local time is shown for the DF90 algorithm described herein (green), the observed shadow (red), the ArcGIS Solar Analyst shadow (black), the Marsh 2012 algorithm (blue), and the white region is the region not

covered by the photograph. The DF90 implementation agrees quite well with observed shadow locations and a sensitivity test (not shown) shows improved agreement with increasingly small triangles. The performance of the other two shadowing algorithms is detailed in Marsh et al. (2012). In brief, the high resolution SolarAnalyst performed the best, however at the cost of multiple hours of runtime. The Marsh 2012 algorithm over predicted the shadow on the right hand side of the domain

whereas DF90 under predicted the shadow delineation. In both cases, the TIN algorithms had a few incorrect shadow classifications on the upper slope as a result of the reduced spatial resolution causing a small false positive. The triangular shaped bumps along the shadow line are from the unstructured triangular mesh elements.

## 6.3 Leave one out validation

The leave one out validation is shown in Figure 9 for Vista View (top row), Upper Clearing (middle row), and Fisera Ridge

(bottom row). The dashed line is the 1:1 line, and the solid black line is a linear regression line of best fit. The $r^2$ value for this fit is shown in the bottom right corner. Due to significant over-plotting of the data points, the values have been binned into 100 hex-bins and coloured using the log of the normalized per-bin count. Hex-bins divide the x-y plane into 6-sided bins and counts values in these bins. The hexes avoid the visual artefacts that can occur with square bins. Grey values are bins that have a normalized count of less than 0.01. Because of the significant number of low and zero values in the shortwave

and precipitation timeseries, this resulted in the per-bin colouring being difficult to read. Values of ISWR < 50 W m$^{-2}$ and p < 1 mm were removed for the colouring. Please note that these data *were not removed* for the linear fit, $r^2$, MBE, or RMSE metrics.

Temperature was well predicted at all sites with $r^2$ values of 0.99, 0.97, and 0.92 for Vista View, Upper Clearing, and Fisera Ridge respectively. Both mid-elevation sites were better predicted than the high elevation (Fisera Ridge) site. The majority

of the data lies close to the 1:1 line. Upper Clearing had a warm bias (MB=1.11 ºC), whereas Fisera Ridge had a cold bias (MB=-0.37 ºC). Less spread was observed in the summer months (not shown), matching the results of Cullen and Marshall (2011). Relative humidity was the most poorly predicted variable. Vista View was the most accurately predicted ($r^2$=0.9) with a slight (1.09%) positive bias. Upper Clearing had more spread with a distinct negative bias (-6.2%) and decreased $r^2$ (0.76). Fisera Ridge was the most poorly predicted ($r^2$=0.55, MB=6.12%). A separate analysis that grouped the data into

winter and summer periods (not shown) showed improved results and less spread during the summer months, especially for Fisera Ridge; this summer period had: $r^2$=0.7, MB=7.84%, RMSE=15.32%. Due to the proximity to vegetation, summer evapotranspiration may result in less temporal variability, dampening the responses. The interpolation methods assume a free-atmosphere, and thus do not capture these canopy interactions. During the winter months, the observed RH is predominately dominated by synoptic scale forcing (Cullen and Marshall, 2011) and may be influenced by the sublimation

of intercepted snow in the canopy (Pomeroy et al., 2012) which are not captured by this interpolation. The Fisera Ridge data has had substantial infilling for the RH variable (Fang et al., 2019), and the poor fit of CHM to these infilled data may be as a result of the infilled data using a higher elevation, exposed ridge, that may not be representative of Fisera Ridge.

Shortwave irradiance is generally well captured, although Fisera Ridge has a larger negative bias (-15.79 W m$^{-2}$) than the other two sites. Precipitation at Vista View and Upper Clearing was well predicted, and Fisera Ridge is again the least well predicted.

## 6.4 Parallel scaling

Shown in Figure 10 are the scaling results for 1, 2, 4, 6, 8, 16, and 32 cores. Good scaling is observed with a 1.97x speedup with 2 cores, 7.23x speedup with 8 cores, a 12.3x speedup with 16 cores, and a 20.5x speedup with 32 cores. A sub-linear scaling is expected due to the mixing of domain and data parallel modules. As most compute nodes are approximately 32 cores, this shows good per-node scaling and thus demonstrates motivation for moving towards a distributed memory model, such as MPI.

**7 Outlook**

As described above, this manuscript outlines the goals of the CHM framework and the motivation for its development. The process representations described herein and currently available in CHM are primarily surface processes. A key component for future development is the addition of the full hydrological cycle to CHM. The use of an irregular geometry for the surface discretization somewhat complicates surface and sub-surface computations of lateral mass and energy fluxes.
Although an open research topic as to how best incorporate these fluxes in CHM, there are examples of how triangular mesh elements have been used in other models. This section outlines some of the techniques used in these models, and how they might be incorporated in CHM.

There are benefits to the improved representation of the irregular geometry for various flow and routing algorithms. If line segments (i.e., triangle edges) are used to represent river and stream channels (along with a sub-grid in-channel flow
parameterization), the non-uniform spacing and the unstructured nature of the triangle vertices can more readily represent meandering features than structured meshes (Tucker, 2001). Simple overland flow-routing methods on unstructured meshes can be easily derived to be analogous to the commonly used grid-based flow methods (Tucker, 2001), e.g., D-8 (O'Callaghan and Mark, 1984) and D-inf (Tarboton, 1997). When simulating the landscape evolution of river channels using unstructured and structured meshes, the directional constraints of a maximum of 8 directions of a structured mesh have been
identified as causing non-natural channel evolution (Braun and Sambridge, 1997). More complex PDE formulations, such as the shallow water equations, can be discretized on the unstructured mesh. However, the numerical formulations require use of more sophisticated numerical methods such as the finite element or finite volume approaches. The use of these numerical techniques on triangular meshes is common when estimating shallow water flows (Hagen et al., 2002), and has been used with success in hydrological models (Kumar et al., 2009; Qu and Duffy, 2007).

The vertical coupling of the surface flow with subsurface flows requires subsurface extension of the surface mesh. An approach is to use vertically extruded triangles, forming 3D prisms of some given height that can be stacked vertically. These prisms can then be used to discretize variably saturated flow calculations, such as the Richards equation, and can optionally simulate lateral subsurface flows for a full 3D model (Hopp et al., 2016; Kumar et al., 2009; Qu and Duffy, 2007). The vertical stacking of prisms is currently being used for the above ground discretization in an in-development blowing snow model for use in CHM (Marsh et al., 2019). Alternatively, vertical 1D models of variably saturated flow can be used with various lateral assumptions (Hopp et al., 2016). Thus, there are various possibilities for the inclusion in CHM, ranging from 1D models with no lateral or surface coupling to fully coupled 3D surface-subsurface flows.

One avenue of future research should examine multi-mesh approaches. Depending on the surface and subsurface geology, refinement of the sub-surface mesh based on surface characteristics may be inappropriate. A possible approach would result in the generation of one or more subsurface meshes, refined as appropriate to represent the heterogeneity in subsurface properties. This mesh would then be coupled to the surface mesh through various approaches such as: a gradation in triangles linking the subsurface and surface meshes; a nested mesh approach; or via interpolation between the meshes. All approaches involve various technical challenges as well as accuracy tradeoffs that have not been intensively explored. Full investigation of the merit of multi-mesh approaches must be done, however it likely presents an elegant solution to incorporating heterogeneity where appropriate across multiple processes and scales.

## 8 Conclusion

Simulations of hydrological phenomena are increasingly important for management and prediction of the hydrological cycle under anthropogenic climate change impacts and cold regions are some of the most sensitive regions to these impacts. Spatially distributed models are generally thought to produce improved predictions in cold regions when spatially explicit prognostic variables are required, however substantial challenges including initial conditions, boundary conditions, parameterizations, and computational costs all conspire to limit their applicability. Despite this, hyper-resolution models are increasingly being applied for water management and design decisions. There is a significant opportunity for next-generation models to address challenges in existing models such as the seamless prediction at various spatial and temporal scales, utilization of hyper-resolution data obtained by new remote sensing platforms, quantify structural uncertainty in distributed models, and utilization of modern high-performance computing infrastructure.

In this manuscript a new modelling framework, the Canadian Hydrological Model (CHM), was presented as a first step towards these goals. Key features of CHM include the ability to capture spatial heterogeneity in an efficient manner; to include multiple process representations; to be able to change, remove, and decouple hydrological process algorithms; to work both at a point and spatially distributed; the ability to scale to multiple spatial extents and scale; and to utilize a variety of forcing fields (boundary and initial conditions). The efficient representation of spatial heterogeneity is due to the use of

unstructured, variable resolution triangular meshes. These can represent key landscape heterogeneities such as vegetation and topography with 50% to 95% fewer computational elements versus a fixed resolution mesh.

To demonstrate and test cold regions operations, two snowpack models, Snobal and SNOWPACK, were compared at a point scale in a mountain clearing. Both models performed well, and demonstrated skill in simulating SWE. Although the irregular geometry of a triangular mesh can complicate application of raster-derived methods, there are various mechanisms in CHM to facilitate the adaptation. A new unstructured mesh implementation of the well-known Dozier and Frew (1990) shadowing algorithm was derived to demonstrate adaptation of raster-based algorithms and the use of these mechanisms in CHM. This method performed well compared to existing high-resolution raster-based algorithms (SolarAnalyst) and other unstructured mesh shadowing algorithms. A leave one out validation was done for the meteorological processes and these results showed a high degree of accuracy in the spatial interpolation of meteorological forcing in CHM. Air temperature was the most accurately predicted forcing variable ($r^2$=0.9) and relative humidity was the most poor ($r^2$=0.5 to 0.7). Lastly, a parallel computation scaling test demonstrated a good but sub-linear scaling with number of CPUs, and demonstrated a need for increased parallelism efficiency via distributed memory models, such as MPI.

In summary, CHM is a first step towards a variable resolution explicitly distributed model with a focus for application where cold-region processes play a role in hydrology. Although it remains a work in progress and only snow accumulation and surface meteorology processes are currently implemented, CHM will ultimately include the entirety of the hydrological cycle. The inclusion of irregular geometries is not a significantly problematic aspect for computations of lateral mass and energy exchanges, and other models have used these geometries without issue. However, novel use of multi-mesh approaches to couple various meshes that have been refined for surface and sub-surface optimization are likely a way forward for including increased explicit heterogeneity with a lower computational burden.

## 9 Author contributions

C. Marsh: Initial idea, coding, analysis, manuscript preparation

J. Pomeroy: Idea refinement, analysis refinement, field data experiment design, manuscript revision

H. Wheater: Idea refinement, manuscript revision

## 10 Competing interests

The authors declare that they have no conflict of interest.

## 11 Acknowledgments

The authors would like to acknowledge the financial support of the Canada Excellence and Canada Research Chairs programmes, the Natural Sciences and Engineering Research Council of Canada's Discovery Grants, Alexander Graham Bell Scholarships and Changing Cold Regions Network, Alberta Innovates, and the Canada First Research Excellence Fund supported Global Water Futures programme. Marmot Creek logistical support from the Nakiska Ski Resort and fieldwork support from Michael Solohub, Xing Fang, May Guan, Angus Duncan, Greg Galloway and many others are gratefully noted.

## 12 Code availability

The code for the Canadian Hydrological Model is open-source under the GPLv3 license and is available at https://github.com/Chrismarsh/CHM and archived with DOI https://zenodo.org/record/3554563. The mesh generation software, Mesher, is open-source under the GPLv3 license. It is available at https://github.com/Chrismarsh/mesher

## 13 Figures

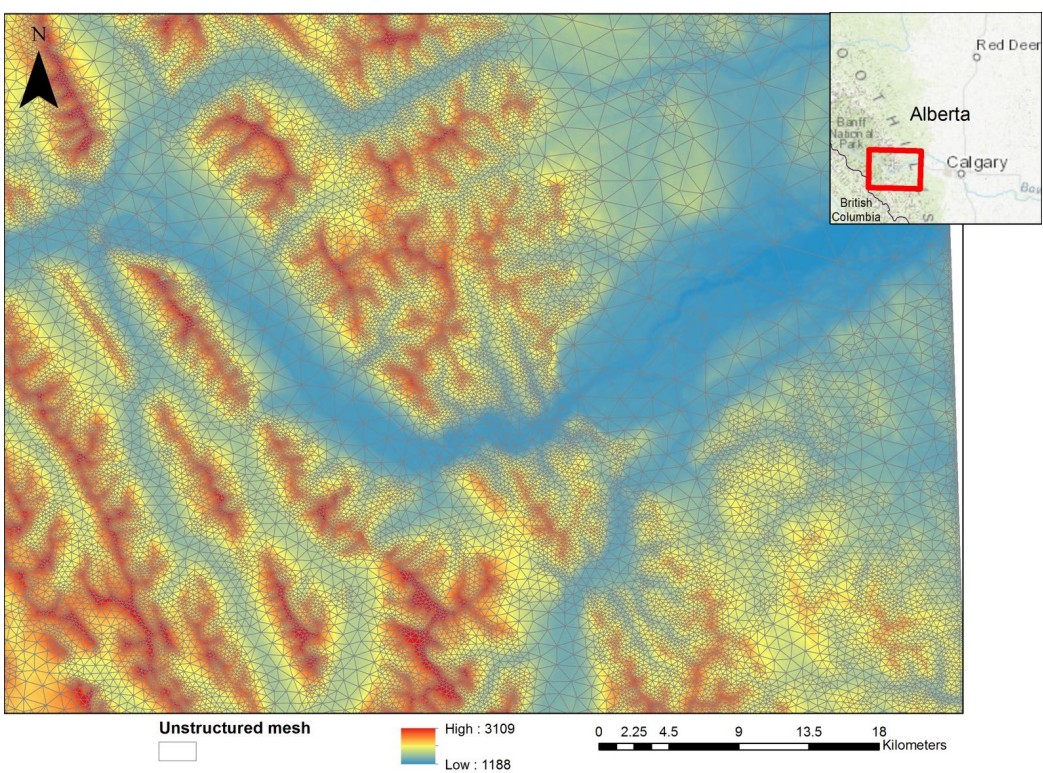

Figure 1: Example of variable resolution triangulation mesh as produced by Mesher for the Bow River Basin west of Calgary in the Canadian Rocky Mountains and foothills. The triangular edges are shown as grey lines overlain on the original DEM.

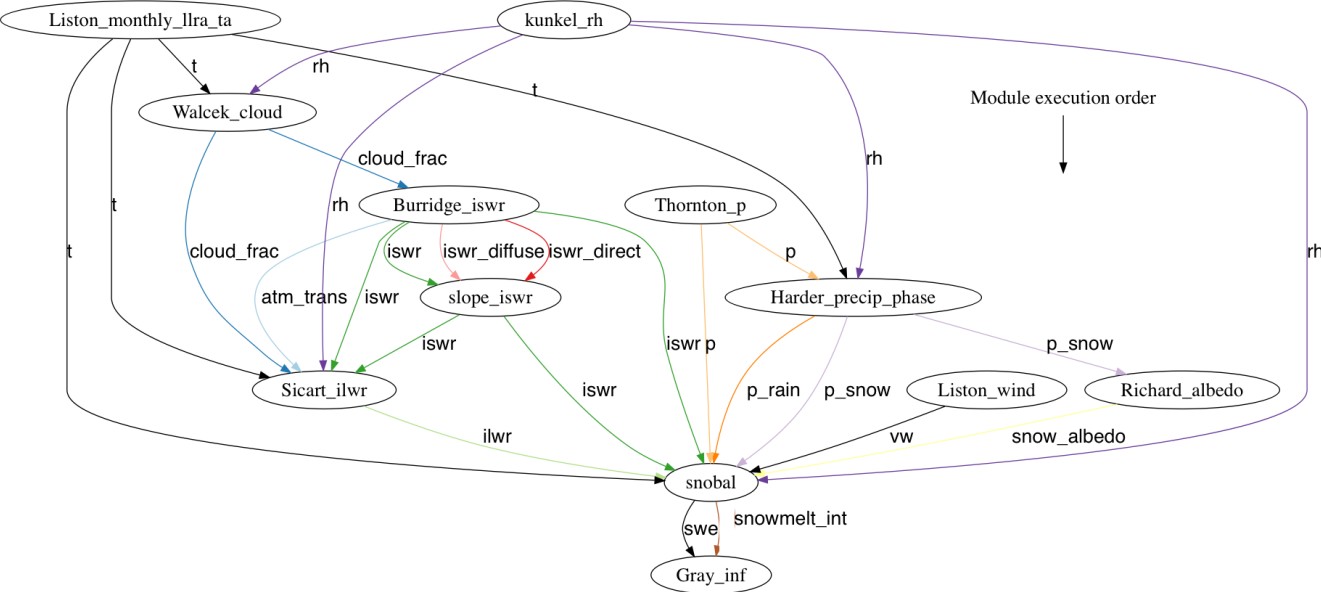

Figure 2: Directed acyclic graph showing module dependencies. Lines point to the module that requires the listed dependency. In this example, a snowcover model, Snobal, is being driven by meteorology in order to drive a frozen soil infiltration model (Gray_inf)

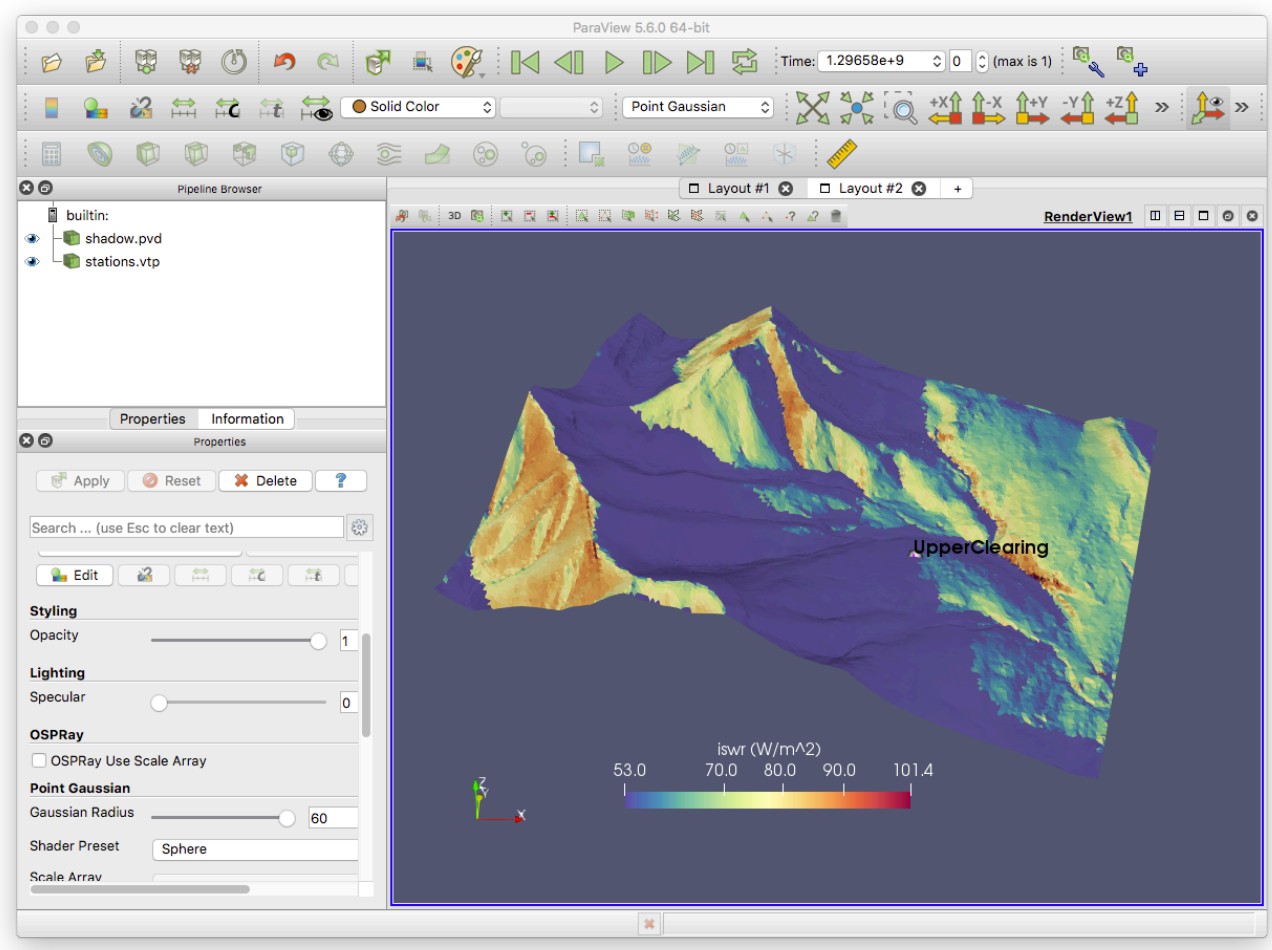

Figure 3: Output from CHM is in the ParaView format, allowing for timeseries analysis and full 3D visualization in ParaView. Shown is shadowing over Marmot Creek Research Basin, Alberta, Canada.

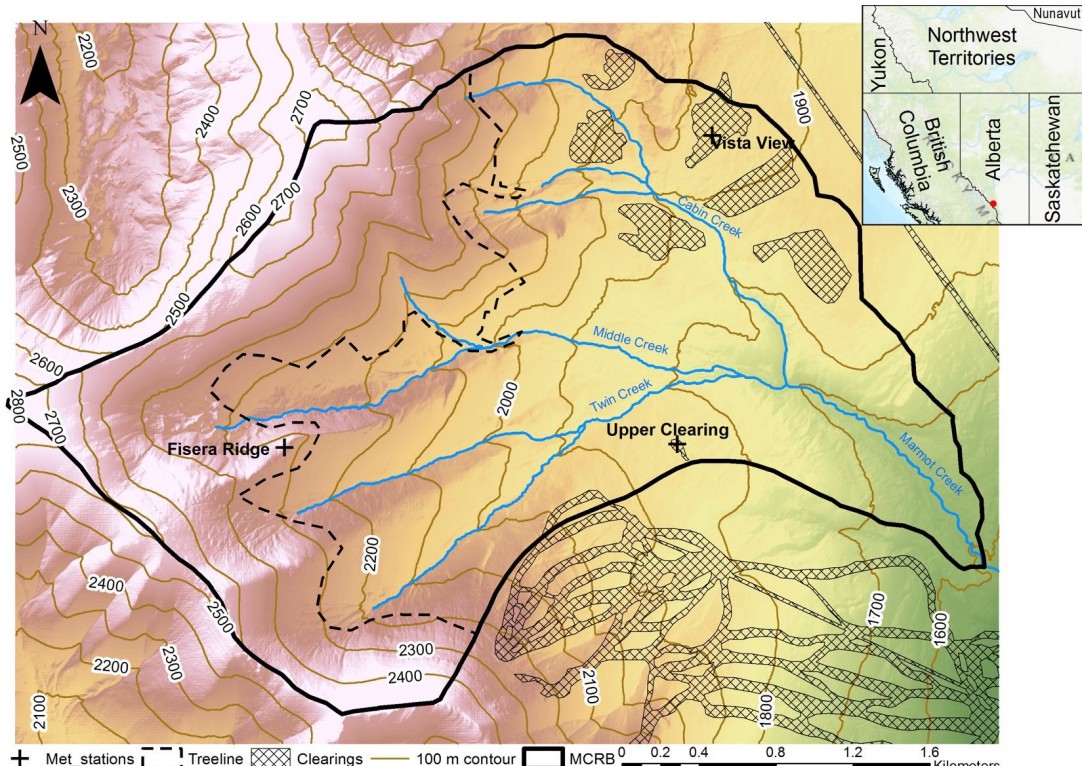

Figure 4: Marmot Creek Research Basin, Kananaskis Valley, Alberta in the Canadian Rocky Mountains. The basin outline is given as solid black, 100 m contour lines shown in brown, stream channels shown in blue, and man-made clearings shown as hatched areas. The meteorological stations used for this study are shown as crosses. The southern-most set of clearings are ski runs in the Nakiska Ski Resort.)

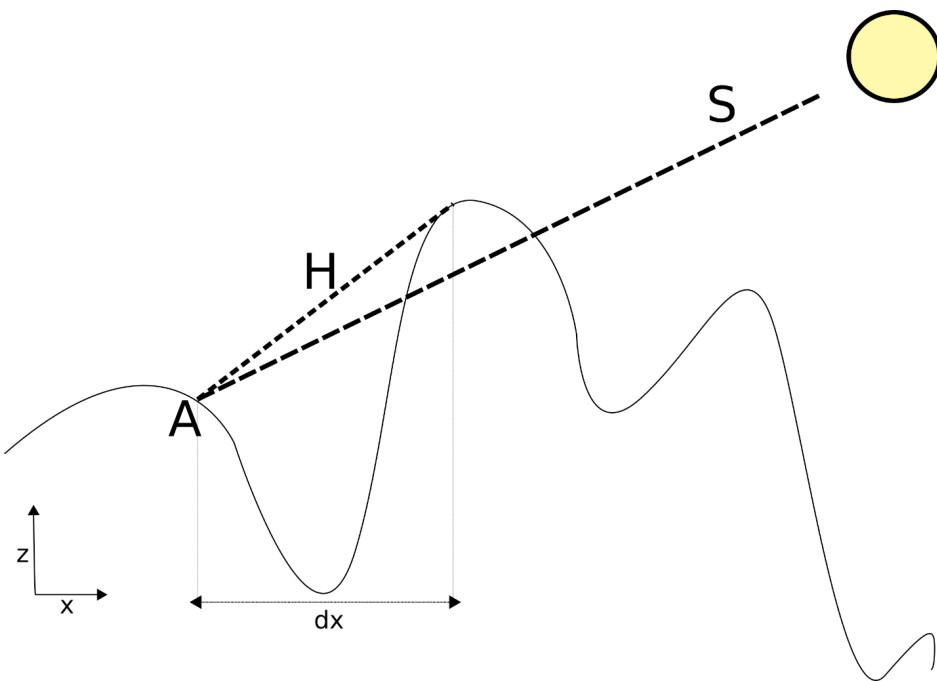

Figure 5: Dozier and Frew (1990) horizon shadowing algorithm. For observer *A*, a search along the azimuth that corresponds to the solar vector S is performed such that if the slope of *H* is greater than that of *S*, *A* is in shadow.

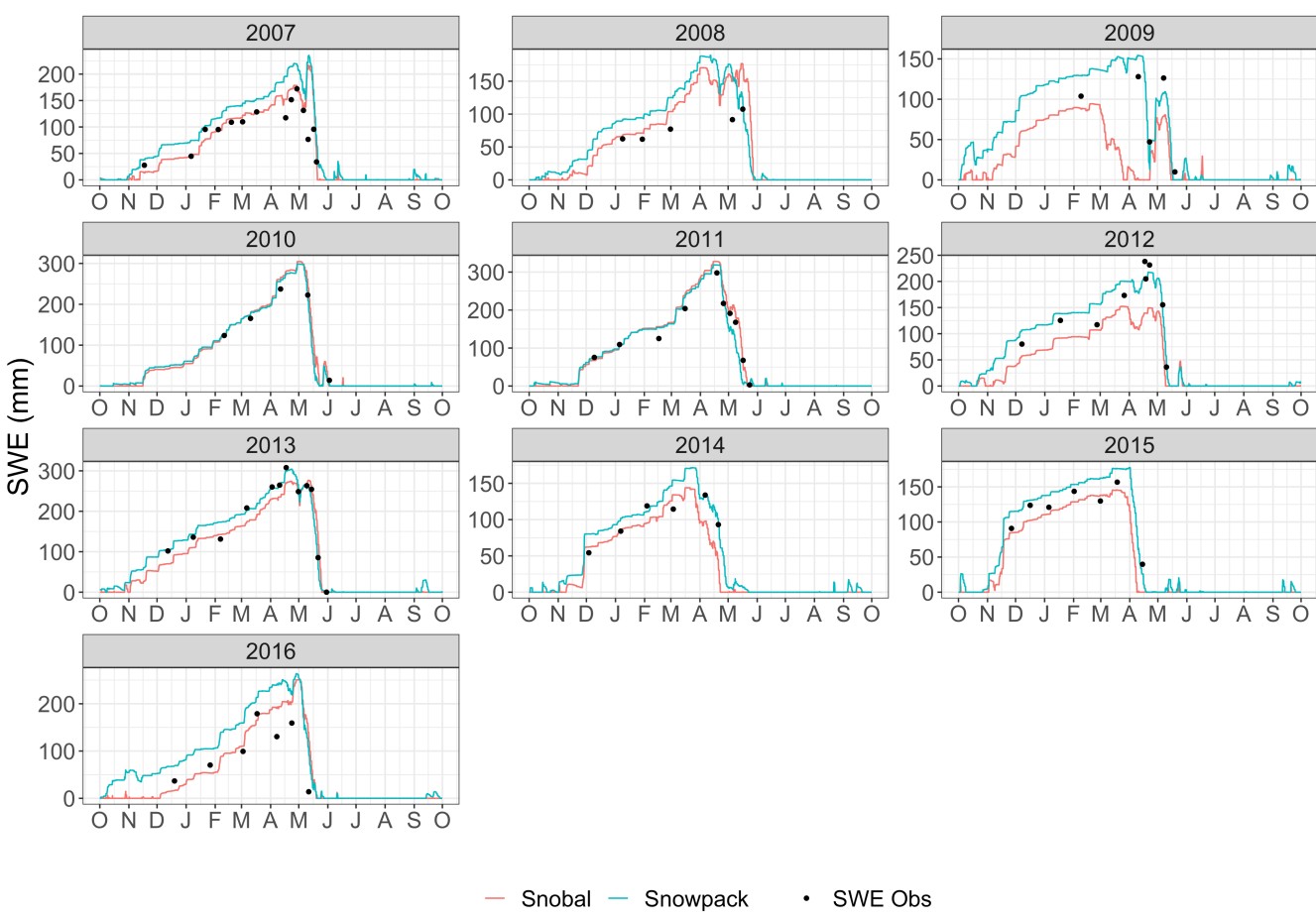

Figure 6: Comparison of Snobal (red) and SNOWPACK (blue) run as a point simulation within CHM for the Upper Clearing site at Marmot Creek Research Basin for 10 water years. Manual snow course observations are shown as black dots.

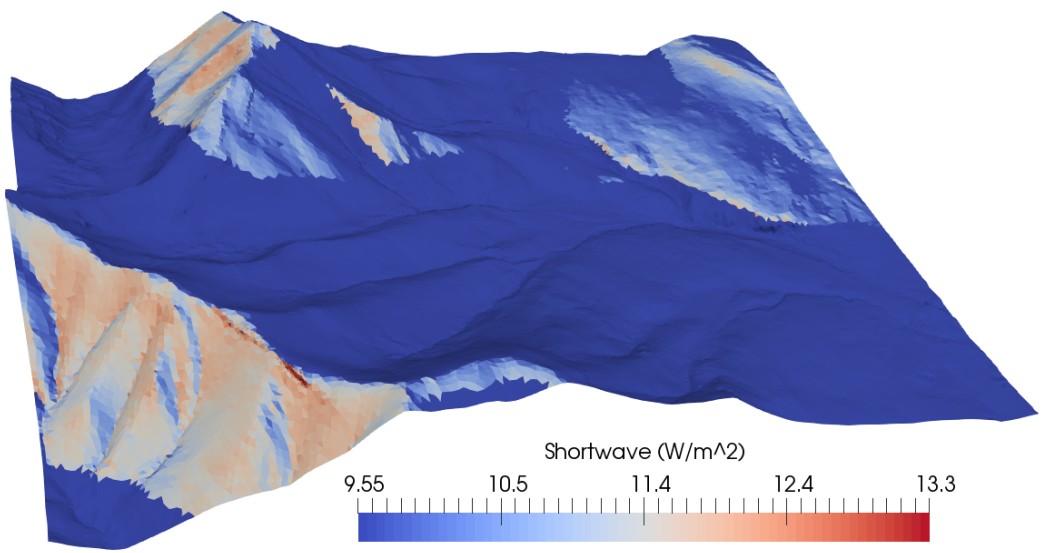

Figure 7: Incoming shortwave radiation for the Marmot Creek Research Basin for 2011-02-01 17:00 local time. The shadowing algorithm of Dozier and Frew (1990) (DF90) has been implemented on the unstructured mesh. Uniform dark blue are shadowed areas.

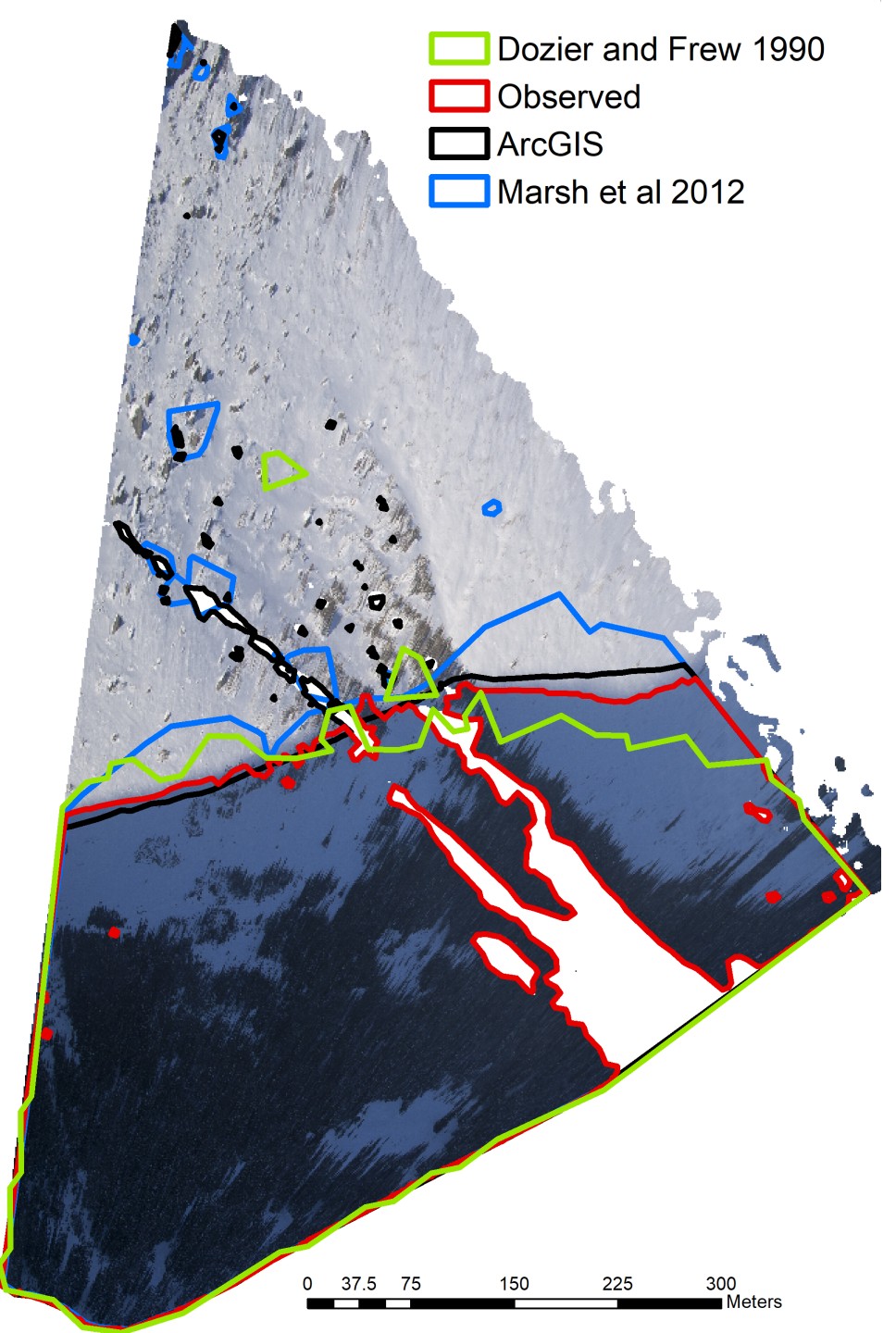

Figure 8: This shows an orthorectified terrestrial photo of a shadow passing over Mt. Collembola from Fisera Ridge – details are found in Marsh et al. ([2012]). The location of the shadowed region for 2011-02-01 17:00 local time is shown for the DF90 algorithm described herein (green), the observed shadow (red), the ArcGIS implementation for a 1m x 1m LiDAR raster (black), and for the Marsh et al. ([2012]) algorithm (blue).

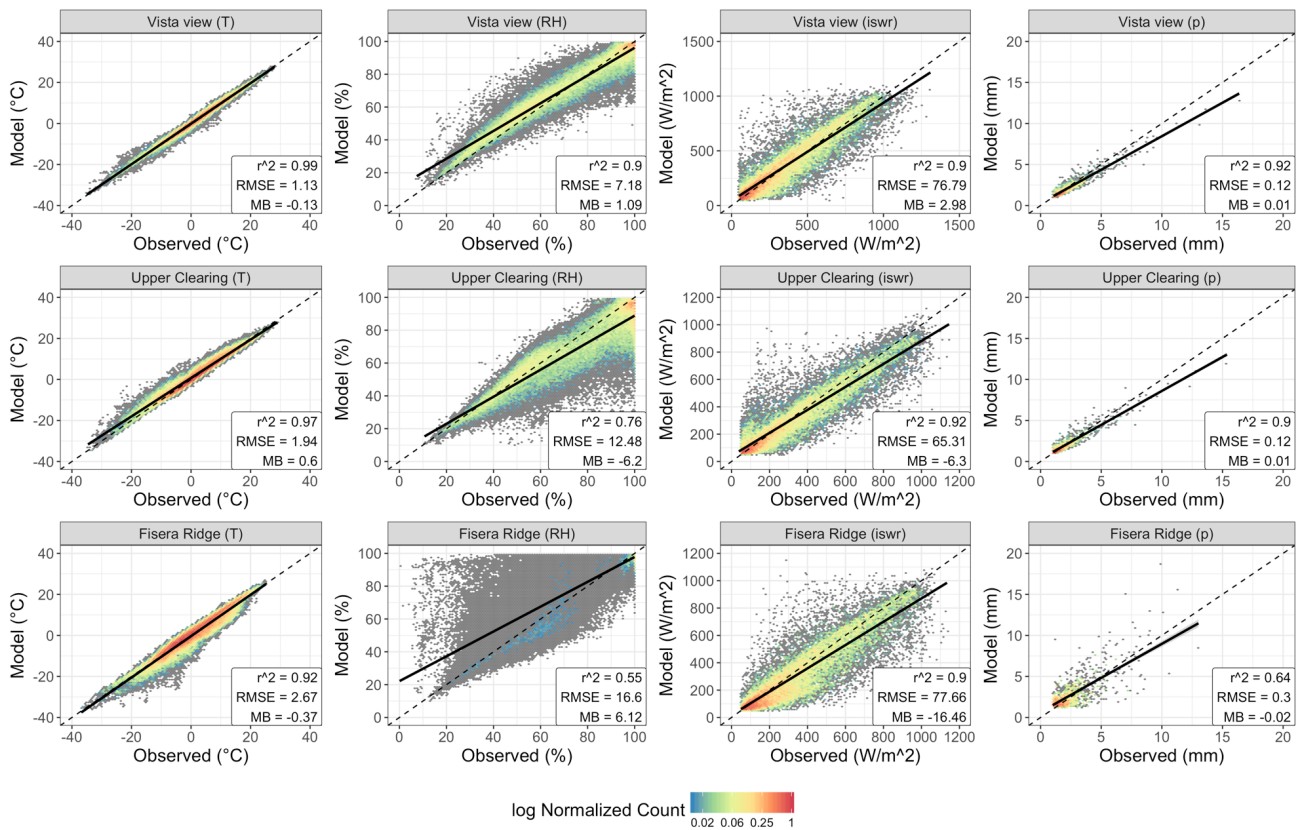

Figure 9: Leave one out analysis for Vista View (top row), Upper Clearing (middle), and Fisera Ridge (bottom). The values have been binned into 100 hex-bins and coloured using the log of the normalized per-bin count. Grey values are bins that have a normalized count of less than 0.01.

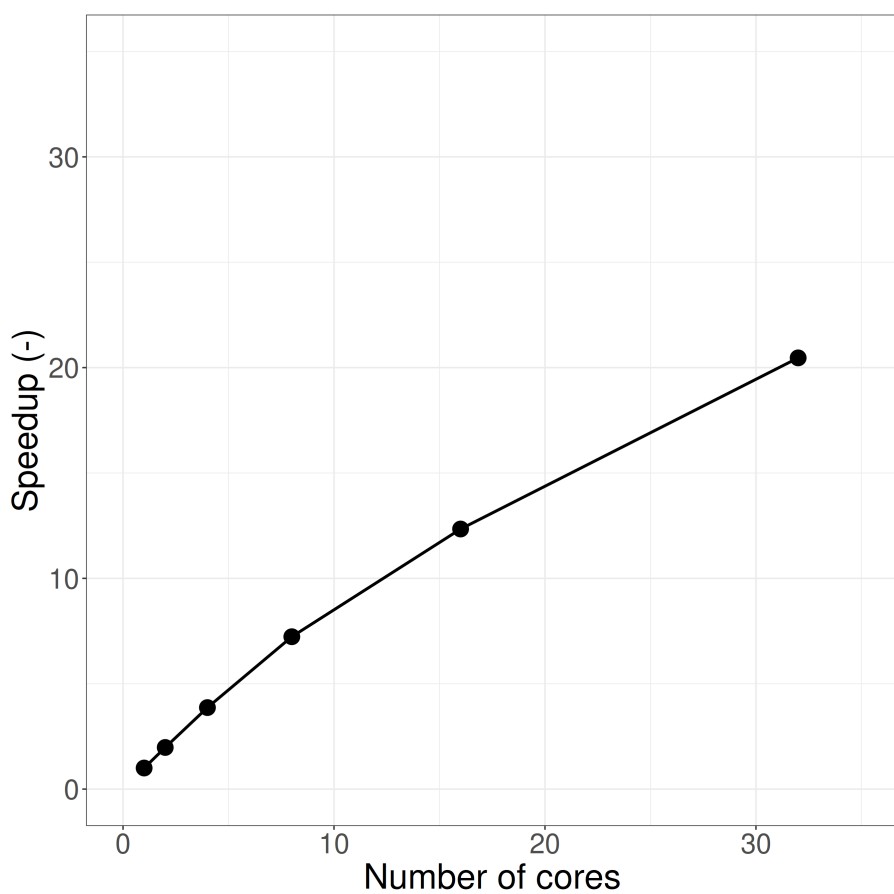

Figure 10: Speedup for a ≈ 100,000 triangle mesh using 1, 2, 4, 6, 8, 16, and 32 cores.

**14 Tables**

Table 1: Cold regions surface process representations currently available in CHM

| Process | Type/name |
| --- | --- |
| Canopy | Open/forest (interception, sublimation, unloading, sub-canopy radiation, turbulent transfer) (Ellis et al., 2010; Pomeroy et al., 1998b) |
| Snowpack | 2-layer Snobal (Marks et al., 1999); Multi-layer SNOWPACK (Bartelt and Lehning, 2002); Various albedo e.g., CLASS (Verseghy, 1991) |

| | |
|---|---|
| Soil | Frozen soil infiltration (Gray et al., 2001) |
| Snow mass redistribution | PBSM3D (Marsh et al. (2019), in review); Snowslide (Bernhardt and Schulz, 2010) |

Table 2: List of available meteorology interpolants.

| Variable | Type |
| --- | --- |
| Air temperature | Linear lapse rates (measured, seasonal, constant, neutral stability) (Cullen and Marshall, 2011; Dodson and Marks, 1997; Kunkel, 1989) |
| Relative humidity | Linear lapse rates (measured, seasonal, constant)(Kunkel, 1989) |
| Horizontal wind | Topographic curvature (Liston and Elder, 2006); Mason-Sykes (Mason and Sykes, 1979); Uniform wind |
| Precipitation | Elevation based lapse (Thornton et al., 1997) |
| Precipitation Phase | Linear; Psychometric (Harder and Pomeroy, 2013); Threshold |
| Solar radiation | Terrain shadows (Dozier and Frew, 1990; Marsh et al., 2012); Clear sky transmittance (Burridge and Gadd, 1975); Transmittance from observations; Cloud fraction estimates (Walcek, 1994); Direct/diffuse splitting (Iqbal, 1980) |
| Longwave | T, RH based (Sicart et al., 2006); Constant (Marty et al., 2002) |

Table 3: Root mean squared error (RMSE [mm]) and Mean bias (MB [mm]) for the SNOWPACK and Snobal models at the Upper Clearing site, for each water year.

| Year | Snobal RMSE (mm) | SNOWPACK RMSE (mm) | Snobal MB (mm) | SNOWPACK MB (mm) |
|------|------------------|--------------------|-----------------|--------------------|
| 2007 | 40.7 | 56.73 | 18.24 | 45.56 |
| 2008 | 42.47 | 42.42 | 33.65 | 38.15 |
| 2009 | 65.07 | 23.22 | -49.54 | -1.49 |
| 2010 | 20.55 | 11.24 | 10.81 | 3.94 |
| 2011 | 24.18 | 32.43 | 15.07 | -7.578 |
| 2012 | 57.12 | 22.26 | -49.05 | 6.92 |
| 2013 | 27.21 | 15.51 | -9.591 | -0.05 |
| 2014 | 28.81 | 21.12 | -13.6 | 13.69 |
| 2015 | 19.19 | 19.41 | -13.54 | 13.84 |
| 2016 | 55.55 | 66.87 | 27.08 | 60.7 |

Table 4: Root mean squared error (RMSE [mm]) and mean bias (MB [mm]) errors averaged over all years.

| Model | RMSE (mm) | MB(mm) |
|---|---|---|
| Snobal | 38 | -3 |
| SNOWPACK | 31 | 17 |

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
