# Peer review of "The Canadian Hydrological Model (CHM) v1.0: A multi-scale, multi-extent, variable-complexity hydrological model -- Design and overview"

_Geoscientific Model Development, 2019_

## Short Comment (SC1) · 17 May 2019

Dear authors,

in my role as Executive editor of GMD, I would like to bring to your attention our Editorial version 1.1: http://www.geosci-model-dev.net/8/3487/2015/gmd-8-3487-2015.html This highlights some requirements of papers published in GMD, which is also available on the GMD website in the 'Manuscript Types' section: http://www.geoscientific-model-development.net/submission/manuscript_types.html

In particular, please note that for your paper, the following requirement has not been met in the Discussions paper:

- "The main paper must give the model name and version number (or other unique identifier) in the title."

Please provide the version number of CHM in the title of your revised manuscript.

Additionally, please note, that GMD is encouraging authors to provide a persistent access to the exact version of the source code used for the model version presented in the paper. As explained in https://www.geoscientific-model-development.net/about/manuscript_types.html the preferred reference to this release is through the use of a DOI which then can be cited in the paper. For projects in GitHub (such as CHM) a DOI for a released code version can easily be created using Zenodo, see https://guides.github.com/activities/citable-code/ for details.

Yours, Astrid Kerkweg

---

## Referee Comment (RC1) · Ruzica Dadic (Referee) · 27 May 2019

Dear Editor,

I reviewed the submitted manuscript by Marsh et al., and I'm happy to recommend it for publication. The paper was exciting to read. It is relevant to the scientific community and is well written. It introduces a novel modelling framework for hydrological models, which at the same time aims at increasing spatial resolution where the terrain is complex and decrease spatial resolution where the terrain is more homogeneous.

[Figure]

Unstructured mesh models have been used in ice flow models for a while and it's exciting to introduce them to surface mass balance processes of the cryosphere. And while introducing modules into model frameworks is generally standard for many models and it's not exactly a novel approach, I still think that it fits nicely with into the model framework.

Furthermore, the examples that the authors have presented to explain the strengths of the model framework and appropriate and make the technical nature of the paper easier to digest and understand. They also highlight the relevance of the approach. I look forward to more exciting studies (especailly process-based studies) with the CHM.

I only have a few minor comments:

- Figure 3 is not very useful and can be removed from the manuscript. - Figure 4 is poorly readable and the resolution should be increased. - As stated above, the modular approach to models is not exactly new and that could be a bit toned down in the manuscript.

I look forward to the publication of this manuscript.

Sincerely, Ruzica Dadic

---

## Referee Comment (RC2) · Anonymous Referee #2 · 4 Sep 2019

Marsh et al. present an overview of the Canadian Hydrological Model (CHM), a modular modeling framework that specifically aims at cold regions. The manuscript focuses on the main principles of the model rather than the specific process representations (what authors call the "philosophy and design" of the model – line 21 page 1), including how terrain is represented (section 4.2), how parameters for each mesh element are determined (section 4.3), how modules are organized (section 4.4), and how weather inputs like precipitation, temperature, relative humidity, radiation, and wind are distributed (section 4.5). After a discussion on parallelization, point-simulation modules,

and output visualization, authors also present some examples of model usage for Marmot Creek in the Canadian Rocky Mountains.

CHM has the potential to overcome several issues in the current state of the art of hydrologic models in cold regions. I found of particular interest the use of TINs instead of pixels or HRUs to represent surface topography. While I am lacking full expertise on parallelization, designing a model that is natively and efficiently parallelized is also an asset. Having said that, I do have a few remarks on the current version of this manuscript, which I outline below. I therefore recommend the Editor to reconsider this manuscript after some extensive, but still minor revisions.

After reading the title of this manuscript, I was expecting the description of a full hydrologic model. As far as I was able to understand, the current version of CHM comprises weather-distribution modules, snow modules, and canopy-soil modules related to snow (see Table 1). Ultimately, a hydrologic model should solve the water balance, including evapotranspiration, soil storage, groundwater, surface-runoff generation, and importantly streamflow. Authors say that "the CHM will eventually include the entirety of the hydrological cycle", but only "snow accumulation and surface meteorology processes are currently implemented" (line 4 page 13). I of course agree that a "hydrological model" does not necessarily have to simulate all processes in the water budget, and I also understand that CHM is still under development. At the same time I think that the manuscript title, abstract, and Methods should be revised to be more specific on what CHM simulates at this point and what this manuscript is focusing on.

Related to this, and particularly because the paper is intended to be a discussion of the main "philosophy" of the model, I feel like an outlook section discussing how authors are planning to include "the entirety of the hydrological cycle" would be interesting. For example, it would be interesting to discuss how flow routing will be eventually implemented, since TINs do not necessarily obey to surface-runoff directions and may be (at least partially) decoupled from subsurface-flow direction.

I also suggest authors to clearly define some of the wording in the manuscript. For example, authors say that "there are no explicitly distributed, modular cold regions models". What is the exact definition of modular here? For example, ALPINE3D is a spatially distributed cold-region model, and it does (to my knowledge) offer several process representations for specific model components (for example, snow hydrology, metamorphism etc). It can also be coupled with a flow-routing scheme (StreamFlow, see https://models.slf.ch). So, to me, ALPINE3D is an explicitly distributed, modular cold regions model.

Also, what do "multi-scale" and 'multi-extent" mean in this context? TINs are an interesting solution to make spatial discretization computationally more effective, because they refine spatial resolution based on actual topography, but (at least to me) multi-scale models are designed to explicitly address other scale issues besides heterogeneity in surface topography (e.g., multi-scale parameter spaces, see https://doi.org/10.1029/2008WR007327). Some other instances are included in my list of specific comments below.

- Line 10 page 1: maybe "precipitation-runoff" would be better here rather than "rainfall-runoff", since precipitation is not only liquid in cold regions?

- Line 18 page 1: maybe introducing TINs here would be more informative than just saying that the model "captures spatial heterogeneity in the surface discretization in an efficient manner"?

- Introduction: I think this Section could be revised for conciseness and to better streamline the story. For example, most of the caveats mentioned in the first paragraph are then discussed at pages 4 and 5, while the problems with raster-based models are described both at lines 9ff page 4 and then at lines 6ff page 5.

- Line 6ff page 2: among these limitations, I would also mention that there are processes that we are simply unable to measure and thus to model without some kinds of parameter tuning (e.g., groundwater storage is often poorly constrained).
- Line 30 page 3: could you provide examples of these "next-generation data products"? If UAVs are such an example, then moving the discussion from line 22 page 5 to here could clarify the point.

- Line 22 page 7: remove one "in" before Marsh et al. 2018.

- Section 4.4: I would expand this section to include details of the modules that are currently supported and their main parametrizations. Currently, this is briefly done in Table 1 and at lines 4-5 page 13, but Section 4.4 seems the adequate place to do so to me.

- Line 17 page 10: what does "embarrassingly" mean here?

- Line 13 page 12: maybe remove "in the results"? Also, how does the animation view specifically allow for immediate diagnosis of modeling errors? Maybe provide a couple of qualitative examples to make the point?

- Line 3 page 14: I believe SNOWPACK is generally reported in all caps

- Line 15 page 14: to my knowledge, SNOWPACK allows for many other turbulent-flux schemes (see again https://models.slf.ch)

- Line 26 page 14: maybe report reference to Figure 6 here?

- Line 1 & 5 & 8 page 15: why did you choose 1000 m and 10 steps here? Maybe providing some of your experience here may guide future users.

- Line 8 page 16: is 2007 actually 2008 here?

- Line 22 page 16: I would include here more details on how the other parametrizations performed.

- Conclusion: I think the first two paragraphs could be summarized or removed, while I would expand on the last paragraph to (1) explicitly mention the pros and cons of TINs, (2) include some of your results from Section 6, and maybe (3) add details of future

[Figure]

work (see my general comments above)

- Figure 11: maybe reports dots to highlight speedup values for 1, 2, 4, 6, 8, 16, and 32, which are those measured in your sensitivity test?

---

## Author Comment (AC1) · 7 Nov 2019

Dear Dr. Kerkweg,

Thank you for you bringing the version number requirement to our attention. This was an oversight, and the title of the manuscript has been changed to include "v1.0".

---

## Author Response (AR1)

Dear Dr. Kerkweg,

Thank you for you bringing the version number requirement to our attention. This was an oversight, and the title of the manuscript has been changed to include "v1.0".

Dear Dr. Dadic (Reviewer #1),

Thank you for your comments.

Figure 3 is not very useful and can be removed from the manuscript.
This has been removed and the text updated.

Figure 4 is poorly readable and the resolution should be increased.
A new figure has been generated at a higher DPI

As stated above, the modular approach to models is not exactly new and that could be a bit toned down in the manuscript.
Agreed. However, it is an uncommon feature for an explicitly distributed cold region model. In the modular process representation section, "A key feature" has been changed to "a feature" to hopefully tone down the use of modular. Regardless, modular process representation does represent a key design characteristic of CHM and warrants an explanation. Hopefully this is sufficient to address this concern.

Dear Reviewer #2,
Thank you for your review.

>At the same time I think that the manuscript title, abstract, and Methods should be revised to be more specific on what CHM simulates at this point and what this manuscript is focusing on.

The following sentence "Although the CHM will eventually include the entirety of the hydrological cycle, snow accumulation and surface meteorology processes are currently implemented." Has been added to the Design and Overview – Overview section to more explicitly acknowledge this limitation, and to ensure the reader is better prepared for the process representations described. The methodology already states this, so hopefully the addition to the Overview section is sufficient to make this point clearly. Regarding the title, we believe the title is sufficient and that describing the overall design goals of the framework with key cold region processes is a reasonable approach.

>I feel like an outlook section discussing how authors are planning to include "the entirety of the hydrological cycle" would be interesting.

A common question has been how to deal with the irregular geometry with overland and subsurface flows. A new section (Outlook) has been added that describes some other models' approaches to this, including some possible avenues for CHM.

>I also suggest authors to clearly define some of the wording in the manuscript. For example

>Line 10 page 1: maybe "precipitation-runoff" would be better

I would like to keep rainfall-run off as I am referring specifically to the non-cold regions literature.

>Line 18 page 1: maybe introducing TINs here would be more informative

I've added "via variable resolution unstructured meshes" to this line

>Introduction: I think this Section could be revised for conciseness and to better streamline the story.

The introduction has been revised to improve the story. A new opening paragraph to more readily articulate the problem statement has been added, and the other paragraphs have been tweaked and reordered to follow a more logical progression.

>Line 6ff page 2: among these limitations,

The first line has been amended to "substantial heterogeneity and difficulty in observing surface and subsurface parameters and processes" which should better link the details later in the paragraph.

>Line 30 page 3: could you provide examples of these "next-generation data products"?

I've added "such as unmanned aerial vehicle (UAV) imagery (Buhler, et al 2016; Harder, et al 2016; Spence, et al 2016]." To clarify. However, I do want the more detailed description in the paragraph following the list of features a next-gen model should have.

>Line 22 page 7: remove one "in" before Marsh et al. 2018.

This is fixed

>Section 4.4: I would expand this section to include details of the modules that are currently supported and their main parametrizations.

This has been added.

>Line 17 page 10: what does "embarrassingly" mean here?

This is a nomenclature common in computer science and means a type of parallel problem where no communication between the workers (threads, MPI processes, etc) is required. It's the simplest type of parallel problem. The text has been amended to include "-- that is, a problem that does not require any communication between threads"

> Line 13 page 12: maybe remove "in the results"?

Agreed

>Also, how does the animation view specifically allow for immediate diagnosis of modeling errors? Maybe provide a couple of qualitative examples to make the point?

The following has been added: "It also allows for immediate diagnosis of modelling errors, especially if the spatial pattern of an output variable is clearly incorrect. For example, if a coding error resulted in: a patch-work of air temperatures instead of an expectedly smooth gradient with elevation, snowdrifts being formed in locations that were known to be incorrect such as the top of a ridge instead of in the lee, or northern hemisphere north-facing slopes receiving the most shortwave irradiance."

>Line 3 page 14: I believe SNOWPACK is generally reported in all caps

This has been changed throughout

>Line 15 page 14: to my knowledge, SNOWPACK allows for many other turbulent-flux schemes

Yes, for example some Antarctica specific parameterizations. The one used herein is the default, wildly applicable scheme. The text has been amended with "The default Michlmayr, et al (2008) scheme was used herein." To clarify which was used.

>Line 26 page 14: maybe report reference to Figure 6 here?

Added, and the later reference was removed

>Line1&5&8page15: whydidyouchoose1000mand10stepshere? Maybe providing some of your experience here may guide future users.

The following has been added to the 2nd paragraph in the "Raster algorithm adaptation (shadowing)" section: "The guidelines for choosing these search values follows two criteria: 1) the radius should be large enough to cover the distance across a representative valley length distance, such that shadows from mountains across the valley are included; and 2) the step should be about half of a triangle length scale such that steps do not pass over triangles."

>Line 8 page 16: is 2007 actually 2008 here?

Yes, thankyou

> Line 22 page 16: I would include here more details on how the other parametrizations performed.

Although extensively detailed in Marsh 2012, an overview of the results was added to this paragraph.

>Conclusion: I think the first two paragraphs could be summarized or removed

The first two paragraphs have been combined into 1 and a new paragraph summarizing the findings from sec 6 were added.

>Figure 11: maybe reports dots to highlight speedup values

added

**The Canadian Hydrological Model (CHM) v1.0: A multi-scale, multi-extent, variable-complexity hydrological model -- Design and overview**

Christopher B. Marsh[1,2]

5   John W. Pomeroy[1,2]

Howard S. Wheater[2,1]

**Affiliations**: [1] Centre for Hydrology, University of Saskatchewan, Canada.; [2] Global Institute for Water Security, University of Saskatchewan, Canada

**1 — Abstract**

10   Despite debate in the rainfall-runoff hydrology literature about the merits of physics-based and spatially distributed models, substantial work in cold regions hydrology has shown improved predictive capacity by including physics-based process representations, relatively high-resolution semi- and fully-distributed discretizations, and use of physically identifiable parameters that require limited calibration. While there is increasing motivation for modelling at hyper-resolution (< 1 km) and snow-drift resolving scales (~1 m to 100 m), the capabilities of existing cold-region hydrological models are

15   computationally limited at these scales.

Here, a new distributed model, the Canadian Hydrological Model (CHM), is presented. Although designed to be applied generally, it has a focus for application where cold-region processes play a role in hydrology. Key features include the ability to capture spatial heterogeneity in the surface discretization in an efficient manner via variable resolution unstructured meshes; to include multiple process representations; to be able to change, remove, and decouple hydrological process algorithms; to

20   work both at a point and spatially distributed; the ability to scale to multiple spatial extents and scale; and to utilize a variety of forcing fields (boundary and initial conditions). This manuscript focuses on the overall model philosophy and design, and provides a number of cold-region-specific features and examples.

**2 — Key points**

- Novel unstructured mesh discretization allows for reduced computational cost while including spatial heterogeneity.

25   • Ability to modify structure and algorithms within a distributed framework allows for in-depth uncertainty testing.

- Flexible spatial and temporal scales, software abstraction, and robust pre- and post-processing routines allow for incorporating existing code, decreasing development effort.

**3 — Introduction**

Hydrological models are important tools for understanding past  hydrological events,   evaluating anthropogenic impacts on natural systems , and informing water resource and management decisions under contemporary and future climates (DeBeer et al., 2015; Freeze and Harlan, 1969; Milly et al., 2008; Mote et al., 2005; Nazemi et al., 2013; Wheater, 2015). ~~Despite the need for hydrological modelling, predictive capabilities are hampered by significant limitations in our modelling ability due to, for instance, substantial heterogeneity in surface and subsurface parameters (Freeze, 1974), the fact that there is 
[revised manuscript text omitted]
. However, they have unique modelling challenges. Increasing importance is being given to rigorous uncertainty analysis, process representation testing, and multiple hypothesis testing. Spatially distributed models are generally thought to produce improved predictions in cold regions when spatially explicit prognostic variables are required, however substantial challenges including initial conditions, boundary conditions, parameterizations, and computational costs all conspire to limit their applicability. Despite this, hyper-resolution models are increasingly being applied for water management and design decisions.

There is a significant opportunity for next-generation models to address challenges in existing models and adapt the large successes from hydrological modelling. These challenges includesuch as the seamless prediction at various spatial and temporal scales, utilization of hyper-resolution data obtained by new remote sensing platforms, quantify of structural uncertainty in distributed models, and utilization of modern high-performance computing infrastructure.

[revised manuscript text omitted]